# PRLR and CACNA2D1 Impact the Prognosis of Breast Cancer by Regulating Tumor Immunity

**DOI:** 10.3390/jpm12122086

**Published:** 2022-12-19

**Authors:** Jiamin Liang, Yu Deng, Yubi Zhang, Bin Wu, Jing Zhou

**Affiliations:** 1Department of Breast and Thyroid Surgery, Union Hospital, Tongji Medical College, Huazhong University of Science and Technology, 1277 Jiefang Avenue, Wuhan 430022, China; 2Department of Orthopaedics, Union Hospital, Tongji Medical College, Huazhong University of Science and Technology, 1277 Jiefang Avenue, Wuhan 430022, China; 3Department of Breast and Thyroid Surgery, People’s Hospital of Dongxihu District Wuhan City and Union Dongxihu Hospital, Huazhong University of Science and Technology, Wuhan 430040, China

**Keywords:** breast cancer, CeRNA, TCGA, PTEN, PRLR, CACNA2D1, tumor immune

## Abstract

Phosphatase and tensin homolog (PTEN) is one of the highly susceptible genes to breast cancer (BC); however, the role of PTEN-related RNAs in BC remains poorly understood. Understanding the effect of PTEN-related RNAs and their mechanisms may be helpful to clinicians. We screened the differentially expressed RNAs (deRNAs) related to PTEN and established the competitive endogenous RNA (ceRNA) network by integrating several databases. After that, the RNA model, prolactin receptor (PRLR)/calcium voltage-gated channel auxiliary subunit alpha2delta 1 (CACNA2D1), was obtained by KM survival analysis and logistic regression analysis. Finally, mutation, methylation, functional enrichment, and immune correlation were analyzed to explore the roles of these RNAs. Our results showed that PRLR might be harmful to BC, while CACNA2D1 might be beneficial to BC. Furthermore, the abnormal expression of PRLR in BC might result from mutation and hypomethylation, while the aberrant expression of CACNA2D1 might be ascribed to methylation. Mechanistically, PRLR might affect the prognosis of BC by inhibiting the expression of immune checkpoints, while CACNA2D1 might improve the prognosis of BC by increasing the immune cells infiltrating into BC and up-regulating the expression of immune checkpoints. The abnormal expression of PRLR and CACNA2D1 in BC is closely related to the prognosis of BC, and they may serve as targets for the treatment of BC.

## 1. Introduction

In 2020, breast cancer (BC) surpassed lung cancer to become the highest incidence cancer world-wide [1]. While most patients with early-stage BC can be cured by local treatment in combination with systemic therapy, about 30–40% of them progress to the advanced stage, i.e., developing relapse and metastasis. These patients hardly respond to and even develop resistance to currently available drugs, and their five-year survival rate is low, posing a huge challenge to the management of BC. Therefore, it is of great significance to understand the mechanism of development and progression of BC and identify new targets for the treatment of BC to improve its prognosis.

Public databases, such as The Cancer Genome Atlas (TCGA) and Gene Expression Omnibus (GEO), provide clinical data and molecular information to facilitate the research of various cancers. Meanwhile, high-throughput RNA sequencing, gene mutation analysis, methylation analysis, and immune microenvironment analysis have been extensively employed in cancer research [2]. In recent years, researchers have been using these techniques and tools to examine tumor RNA in molecular terms. Their studies have screened a great many potential targets associated with the diagnosis and treatment of tumors. For instance, Huang et al. identified CD247 from TCGA and GEO databases, and Fc gamma receptor Ia (FCGR1A) and transformation/transcription domain associated protein (TRRAP) were used as potential therapeutic targets for the treatment of cholangiocarcinoma [3]. Wang et al. found that lnc-CTSLP8 could act as a potential target for treating metastatic ovarian cancer by searching TCGA and GEO databases [4]. Zhou et al. found that Hsa-circ-0001666 could hinder the progression of colorectal cancer via the miR-576-5p/PCDH10 axis, which may provide a novel approach for the treatment of colorectal cancer [5]. The aforementioned studies theoretically and methodologically paved the way to the further screening of new therapeutic targets for BC.

As a common tumor suppressor gene, phosphatase and tensin homolog (PTEN) inhibits the proliferation of cancer cells by down-regulating the PI3K/AKT/mTOR pathway [6,7], and is implicated in the development and progression of tumors, such as hepatocellular carcinoma [8], gastric cancer [9], prostate cancer [10], osteosarcoma [11], nasopharyngeal carcinoma [12], and BC. At the same time, PTEN, as one of the genes with a high genetic susceptibility to BC, is responsible for an 85% lifetime risk of BC. Multiple clinical trials, such as phase II study (GO29227, LOTUSNCT02162719) and phase III Ipatunity 130 trial (NCT03337724), have suggested that PTEN is clinically important in the treatment of BC [7], but no research investigated the role of PTEN-related RNAs in the development and progression of BC.

On the basis of prior studies, we constructed a competitive endogenous RNA (ceRNA) network related to PTEN in BC, and then analyzed a KM survival, functional enrichment, and PPI network on the RNAs in the network. Then, the obtained RNAs linked to BC prognosis were subjected to logistic regression, thereby establishing a BC prognosis model containing prolactin receptor (PRLR) and calcium voltage-gated channel auxiliary subunit alpha2delta 1 (CACNA2D1). Furthermore, the model was evaluated by an internal test set. Then, a correlation analysis showed that both PRLR and CACNA2D1 were related to PTEN, which might be involved in the pathogenesis of BC. Finally, the potential biological functions of PRLR and CACNA2D1 in BC were examined by functional enrichment analysis, and the possible roles of PRLR and CACNA2D1 in BC were studied by analyzing mutation, methylation, and immune correlation (Figure 1). The study aimed to provide a new target for the treatment of BC, to lower the recurrence and improve the prognosis of advanced BC.

## 2. Materials and Methods

### 2.1. Data Acquisition and Processing

Data on RNA expressions and clinical features of BC samples and adjacent tissues were downloaded from TCGA database (https://portal.gdc.cancer.gov/, accessed on 21 December 2021), and RNA and miRNA sequences were obtained from IlluminaHiSeq_RNASeq and IlluminaHiSeq_miRNASeq sequencing platforms. All data could be freely downloaded. In terms of the median PTEN expression, 1059 BC samples were divided into PTEN^high^ group (*n* = 529) and PTEN^low^ group (*n* = 530). Against the requirements of data integrity, the samples with missing items were excluded. Finally, data of 1000 patients were obtained from BC samples, and the caret package was further randomly allocated to the training set and the test set at a ratio of 7: 3. In addition, the GSE21422 data set was downloaded from GEO database (https://www.ncbi.nlm.nih.gov/gds/, accessed on 17 July 2022) to verify the expression of RNAs in the model.

### 2.2. Identification of deRNAs

The criteria for eligible deRNAs in BC samples and normal samples were as follows: mRNA with a *p* < 0.05 and |logFC| > 0.7, lncRNA with a *p* < 0.05 and |logFC| > 0.5, and miRNA with a *p* < 0.05 and |logFC| > 0.3. Meanwhile, the criteria for eligible deRNAs in PTENhigh and PTENlow were: mRNA with a *p* < 0.05 and |logFC| > 0.5, lncRNA with a *p* < 0.05, and |logFC| > 0.5 and miRNA with a *p* < 0.05 and |logFC| > 0.3. The identified demRNA, delncRNA, and demiRNA were presented by volcano maps generated by gplots package in R software (version 4.2.1, https://cloud.r-project.org/).

### 2.3. Construction of ceRNA Network and PPI Network of BC

Venn diagram analysis was performed by using the online tools jvenn (http://jvenn.toulouse.inra.fr/app/example.html, accessed on 10 April 2022), and the targeted relationships of the identified RNAs were predicted by utilizing miRcode (http://www.mircode.org/, accessed on 30 January 2022), TargetScan (https://www.targetscan.org/vert_71/, accessed on 30 January 2022), and miRTarBase (https://mirtarbase.cuhk.edu.cn/~miRTarBase/miRTarBase_2022/php/index.php, accessed on 30 January 2022) databases. Then, the protein–protein interaction (PPI) network for the resulting mRNAs was constructed by employing the STRING database (string-db.org, accessed on 6 February 2022). Finally, Cytoscape software (version 3.9.1, https://cytoscape.org/download.html) was used to visualize the ceRNA network and PPI network.

### 2.4. Functional Enrichment Analysis

In order to understand the biological functions of the network and mRNA molecules, we used MetaScape (http://metascape.org, accessed on 6 February 2022) to conduct gene ontology (GO) and Kyoto Encyclopedia of Genes and Genomes (KEGG) enrichment analysis of the identified network and important RNA molecules (the top 200 genes related to PRLR and CACNA2D1 in BC). An adjusted *p* < 0.05 was deemed statistically significant.

### 2.5. Survival Analysis and Construction and Verification of Prognosis Model

The survival analysis of PTEN in BC was carried out by using Kaplan–Meier plotter (http://kmplot.com/analysis/, accessed on 3 January 2022). The survival package of R software was employed to assess the KM survival of deRNAs in ceRNA network, and the variables included in the prognosis model were first screened in terms of a *p* < 0.15. Then, the logistic regression analysis was performed to establish the model, and the model was then verified by chi-square analysis, Hosmer–Lemeshow Test, calibration curve, and ROC curve. In addition, the survival analysis of PRLR and CACNA2D1 was re-verified by using the GEPIA2 database (http://gepia2.cancer-pku.cn, accessed on 8 June 2022).

### 2.6. Analysis of RNA Mutation, Methylation, and Expression

The mutation analysis of PTEN, PRLR, and CACNA2D1 was carried out by utilizing cBioPortal for Cancer Genomics (https://www.cbioportal.org/, accessed on 10 March 2022). Then, UALCAN (http://ualcan.path.uab.edu/, accessed on 7 March 2022) and DeatherMeth 2.0 (http://bio-bigdata.hrbmu.edu.cn/diseasemeth/, accessed on 15 March 2022) were used to evaluate the methylation levels of PRLR and CACNA2D1 in BC tissues and normal tissues, and MEXPRESS (https://mexpress.be, accessed on 15 March 2022) was employed for visualization. Finally, a heatmap was drawn by using MethSurv (https://biit.cs.ut.ee/methsurv/, accessed on 15 March 2022). Meanwhile, the expression of RNAs in BC tissues and normal tissues was analyzed by using the UALCAN database and GEPIA 2 database, and the expression of RNA in BC cell lines and at protein level was verified by the Human Protein Atlas (HPA) database (https://www.proteinatlas.org/, accessed on 10 January 2022) [13,14,15].

### 2.7. Immune Infiltration and Immune Checkpoint Analysis

The TIMER database (https://cistrome.shinyapps.io/timer/, accessed on 28 May 2022) was used to assess the relationship between the expression of PRLR and CACNA2D1 in BC and the abundance of tumor immune infiltrating cells, the relationship between tumor immune infiltrating cells and prognosis, and the correlation between PRLR and CACNA2D1 and immune checkpoints.

## 3. Results

### 3.1. Inhibitory Effect and Prognostic Value of PTEN Overexpression in BC Cells

In the HPA database, we found that PTEN expression was down-regulated in BC tissues compared with normal tissues, and the immunohistochemical results also confirmed that PTEN expression was extremely low in BC specimens (Figure 2A,B and Table 1). Furthermore, comparable findings were obtained in the GEPIA2 and UALCAN databases, and the expression of PTEN gradually decreased with the tumor progression (Figure 2C,D), suggesting that PTEN plays a role in BC. The KM curve exhibited that PTEN expression was significantly related to the overall survival (OS) of patients, and patients with high PTEN expression had a more favorable prognosis (Figure 2E).

Then, we examined why PTEN expression was disproportionately low in BC by using cBioPortal database. OncoPrint map showed the distribution of the PTEN gene was altered in TCGA BC samples, and the mutation rate of PTEN in BC was up to 14% (Figure 2F). Moreover, samples with PTEN deleted had a conspicuously low expression, PTEN mRNA expression was up-regulated with the increase of copy number, and there existed a positive correlation (Figure 2G) between them. These findings suggested that decreased PTEN copy number might contribute to the down-regulated PTEN expression in BC. 

### 3.2. Screening of deRNAs (demRNAs, delncRNAs, and demiRNAs)

On the basis of aforementioned analyses, we speculated that PTEN-related RNAs may serve as a potential therapeutic target for BC. A total of 102 normal breast samples and 1059 BC samples were selected from TCGA database. Against the median expression of PTEN in BC samples upon screening (eliminating metastatic and non-cancerous samples), the samples were divided into PTEN^high^ group (529) and PTEN^low^ group (530). First, BC samples and normal samples were screened for deRNAs, with *p* < 0.05 and |log FC| < 0.7 as eligibility criteria of mRNA, *p* < 0.05 and |log FC| < 0.5 as the criteria for lncRNA, and *p* < 0.05 and |log FC| < 0.3 as the criteria for miRNA. In total, 4346 demRNAs, 446 delncRNAs, and 332 demiRNAs were identified from BC samples and normal samples. Second, the deRNAs of PTEN^high^ group and PTEN^low^ group were screened, with *p* < 0.05 and |log FC| < 0.5 as the mRNA cutoff value, *p* < 0.05 and |log FC| < 0.5 as the lncRNA cutoff value, and *p* < 0.05 and |log FC| < 0.3 as the miRNA threshold. Finally, 1329 demRNAs, 109 delncRNAs, and 109 demiRNAs were found from the PTEN^high^ group and PTEN^low^ group (Figure 3).

### 3.3. Construction and Analysis of ceRNA Network Related to PTEN

In order to construct a PTEN-related ceRNA regulatory network in BC, we compared deRNAs of PTEN^high^ and PTEN^low^ groups with those of BC and normal tissues, and finally identified 700 demRNAs, 47 delncRNAs, and 84 demiRNAs, which were displayed by Venn diagrams (Figure 4A). Then, we used miRcode to predict the miRNAs targeted by delncRNAs, and an intersection set between the predicted results and demiRNAs was obtained. We used miRcode, TargetScan, and miRTarBase in combination to predict the mRNAs targeted by demiRNAs, and obtained the intersection between the predicted results and the aforementioned deRNAs, and finally identified a set that included eighty-two mRNAs, two lncRNAs (HOX transcript antisense RNA (HOTAIR) and X inactive specific transcript (XIST)), and five miRNAs (hsa-miR-429, hsa-miR-190b, hsa-miR-375, hsa-miR-107, and hsa-miR-217), thereby forming a ceRNA network, which was visualized by Cytoscape (Figure 4B). Then, we conducted a functional enrichment analysis of mRNAs in the ceRNA network by using Metascape (Figure 4C) and found that mRNAs in the network were principally involved in cell morphogenesis, intercellular adhesion, and head development. Finally, we constructed a PPI network by using the STRING database (Figure 4D) and further examined the relationship among mRNAs in the network (Appendix A) to screen out the hub genes.

### 3.4. Construction and Validation of the Specific BC Prognosis Model

The nodes in PPI network, miRNAs and lncRNAs in ceRNA network were subjected to KM survival analysis. In terms of the criterion of *p* < 0.15, calcium voltage-gated channel subunit alpha1 D (CACNA1D), potassium calcium-activated channel subfamily N member 4 (KCNN4), PRLR, protein kinase cGMP-dependent 1 (PRKG1), has-miR-429, CACNA2D1, NADPH oxidase 4 (NOX4), EPH receptor A3 (EPHA3), glypican 6 (GPC6), HOXA transcript antisense RNA myeloid-specific 1 (HOTAIRM1), and HOTAIR were preliminarily eligible for being included in the prognostic model. Logistic regression analysis of these variables and clinical characteristics (including age, gender, tumor node metastasis (TNM), tumor stage, and prior malignancies) revealed that only distant metastasis, lymph node metastasis, CACNA2D1, and PRLR registered a significant correlation with the prognosis of BC. Then, the four significant variables were used to construct a model, in which the *p* values of all variables were found to be < 0.05 (Table 2). Meanwhile, the significance of the model was verified by Chi-square test (Table 3), and the calibration of the model was verified by the Homer–Lemeshow test, and the *p* value of the training set was found to be 0.67 (> 0.05), indicating that the current data had been fully extracted and the goodness-of-fit of the model was high. Furthermore, the goodness-of-fit of the model was further demonstrated by the calibration curve (Figure 5A). Finally, the discrimination of the model was verified by the receiver-operating characteristic (ROC) curve on the training set, verification set, and total data set (Figure 5B–D), which indicated that the model had good discrimination and prediction effect. On the basis of the model, we predicted that distant metastasis of tumor, lymph node metastasis, and PRLR expression were the risk factors of unfavorable prognosis. With the increase of PRLR expression, the risk of death of BC patients was on the rise. On the contrary, CACNA2D1 was a protective factor in the prognosis of BC. With the up-regulation of CACNA2D1 expression, the risk of death of BC patients dropped decreasingly.

In verifying the correlation between PRLR and CACNA2D1 and PTEN, we found that the expression of PRLR and CACNA2D1 in BC was moderately correlated with PTEN on the basis of TIMER database (*p* < 0.01) (Figure 6). These data showed that we successfully established a correlation model between BC prognosis and PTEN.

### 3.5. Verification of PRLR and CACNA2D1 Expression in BC and the Causes of Abnormal Expression

Against GEO data set GSE21422, we verified the expression of PRLR and CACNA2D1 in breast tissue, and found that PRLR was highly expressed in BC tissue, while CACNA2D1 expression was low in BC tissue (*p* < 0.001), which was consistent with the results of the bioinformatic analysis (Figure 7A and Table 4). As expected, the GEPIA2 database search revealed that with the increase of PRLR expression, the risk of death was also higher in BC patients. On the contrary, with the increase of CACNA2D1 expression, the risk of death of BC patients was lower (Figure 7B).

Subsequently, we explored the possible mechanism of abnormal expression of PRLR and CACNA2D1 in BC. As we know it, mutations and DNA methylation play an important role in the abnormal expression of genes. Therefore, we first searched cBioPortal database and found that the mutation probability of PRLR was 2.2%, and there was a weak correlation between PRLR expression and copy number, indicating that the overexpression of PRLR might be related to mutations. Meanwhile, the mutation probability of CACNA2D1 was 1.6%, and no correlation was found between its expression and copy number. Therefore, we were led to speculate that the abnormal expression of CACNA2D1 might not be linked to mutation (Figure 7C–E).

Furthermore, we examined whether methylation was responsible for the abnormal expression of PRLR and CACNA2D1. By searching UALCAN, we found that the methylation level of PRLR was lower in BC tissue than in normal tissue, and the methylation of CACNA2D1 was higher in BC tissue than in normal tissue (*p* < 0.001) (Figure 8A). DNA methylation could alter RNA expression by inhibiting gene transcription. Therefore, PRLR may elevate its expression in BC tissues by lowering methylation, and CACAN2D1 may reduce its expression in BC tissues by increasing methylation. In order to locate the methylation sites of PRLR and CACNA2D1 in BC, we visualized the sites with MEXPRESS, and found that five methylation sites of PRLR were positively correlated with its expression, and seventeen sites bore negative correlation with its expression (Figure 8B), while sixteen methylation sites of CACNA2D1 were positively correlated with its expression, and twelve sites bore negative correlation with its expression (Figure 8C). Finally, we used MethSur to present the differential methylation region related to PRLR and CACNA2D1 with heatmaps. Interestingly, the methylation site cg02976952 of PRLR was situated in the 5’UTR region and the open-sea (Figure 9A), while the methylation site cg25161868 of CACNA2D1 was situated in the 3’UTR region and the open-sea (Figure 9B).

### 3.6. Functional Enrichment Analysis of PRLR and CACNA2D1

To further understand the biological functions of PRLR and CACNA2D1 in BC, the top 200 genes related to PRLR and CACNA2D1 in BC were retrieved from GEPIA2, and the GO and KEGG pathway enrichment analyses of these genes were performed by using Metascape. It was found that PRLR was mainly implicated in cytoskeleton-dependent intracellular transport, protein catabolism, and the Hedgehog signaling pathway (Figure 10A), while CACNA2D1 was principally involved in myofibril formation, adhesive spots, and muscle structure development (Figure 10B). Prior studies have shown that cytoskeleton-dependent intracellular transport and the cytoskeleton were related to tumor immunity [16,17,18]. All aforementioned results suggested that PRLR and CACNA2D1 might be related to tumor immunity. We then explored the relationship between PRLR, CACNA2D1, and tumor immunity by looking at tumor immune cell infiltration and immune checkpoints.

### 3.7. Correlation between the Expression of PRLR and CACNA2D1 and the Immune Infiltrating Cells in BC

As a part of the tumor immune response, immune cell infiltration is related to the development, progression, and outcome of tumors. We used TIMER to evaluate the effect of immune cell infiltration on the clinical prognosis of BC patients, and found that lower levels of B cells, CD8+ T cells, CD4+ T cells, neutrophils, and dendritic cells were related to a worse prognosis of BC patients, while the expression of macrophages was not significantly correlated to the prognosis of BC (Figure 11A). Then, we examined the relationship between PRLR, CACNA2D1, and immune infiltrating cells in BC, and found that PRLR expression was not associated with the infiltration level of B cells, CD4+ T cells, neutrophils, and dendritic cells, but was tenuously related to the infiltration level of CD8+ T cells (Figure 11B), indicating that immune cell infiltration may not be culpable for the unfavorable prognosis in BC patients with PRLR over-expression. The expression of CACNA2D1 was positively correlated with the infiltration levels of B cells, CD8+ T cells, CD4+ T cells, neutrophils, and dendritic cells (Figure 11C). In summary, these results suggested that the low expression of CACNA2D1 in BC may affect the clinical prognosis of BC patients by down-regulating the abundance of tumor immune infiltrating cells.

### 3.8. Correlation between PRLR, CACNA2D1, and Immune Checkpoints

The expression of immune checkpoints was also found to be intimately related to the prognosis of a tumor [19]. Programmed cell death 1 (PDCD1), programmed cell death 1 ligand 2 (PDCD1LG2), cytotoxic T-lymphocyte associated protein 4 (CTLA4), lymphocyte activating 3 (LAG3), hepatitis A virus cellular receptor 2 (HAVCR2), T cell immunoreceptor with Ig and ITIM domains (TIGIT), V-set immunoregulatory receptor (C10ORF54), and B and T lymphocyte associated (BTLA) are common immune checkpoints targeted by tumor immunity research. Therefore, we analyzed the relationship between the above-mentioned immune checkpoints and the clinical prognosis of BC patients by searching the GEPIA2 database, and found that PDCD1, CTLA4, TIGIT, and BTLA were closely linked to the prognosis of BC patients. BC patients with high expression of these immune checkpoints had a favorable prognosis, while PDCD1LG2, LAG3, HAVCR2, and C10ORF54 bore no relationship with the prognosis of BC patients (Figure 12A). Then, we analyzed the co-expression relationship between PRLR, CACNA2D1, and PDCD1, CTLA4, TIGIT, and BTLA, and found that PRLR was negatively correlated with PDCD1, CTLA4, TIGIT, and BTLA (Figure 12B). These results suggested that the high expression of PRLR in BC may impact the prognosis of BC patients by inhibiting the expression of PDCD1, CTLA4, TIGIT, and BTLA. Furthermore, CACNA2D1 was positively correlated with the expression of CTLA4, TIGIT, and BTLA, but not with the expression of PDCD1 (Figure 12C), indicating that immune checkpoints may also be held responsible for the poor prognosis of BC patients with low expression of CACNA2D1.

## 4. Discussion

Globally, BC is the highest-incidence cancer world-wide and the leading cause of cancer-related deaths in women [20]. While various treatment strategies are available for BC, some patients, especially those with advanced or metastatic BC, fail to respond to the treatments and end up with poor prognosis. In fact, how to improve the survival rate and quality of life of these patients has become a subject of active studies. In recent years, bioinformatic analysis has provided a new direction for finding potential targets for cancer treatment, and achieved certain curative effects in clinical practice. These developments provided a theoretical basis and methodological support for screening new therapeutic targets for BC in our study.

PTEN can regulate the PI3K–AKT–mTOR signaling pathway to control a wide array of biological processes, including cell survival, migration, proliferation, and metabolism, and its abnormal expression may lead to cancer susceptibility and tumor progression [21]. In this study, survival analysis, IHC, and copy number variation analysis of PTEN were carried out on the basis of the HPA database and cBioPortal database. These results demonstrated that PTEN expression was related to the prognosis of BC patients. However, up till now, the relationship between PTEN-related RNAs and BC has not been studied. Therefore, we attempted to establish a PTEN-related ceRNA network in BC and explored the relationship between RNAs in the network and BC prognosis.

First, a PTEN-related ceRNA network composed of eighty-two mRNAs, two lncRNAs, and five miRNAs was constructed in this study based on TCGA database. Enrichment analysis showed that the biological functions of this network mainly involved the cell morphogenesis and cell adhesion. Then, PPI network analysis, KM survival analysis, and logistic regression analysis exhibited that PRLR and CACNA2D1 were related to the prognosis of BC. Furthermore, we conducted several analyses on PRLR and CACNA2D1, and found that PRLR may be overexpressed in BC tissues through mutation and hypomethylation, thus affecting the prognosis of BC patients. Meanwhile, the expression of CACNA2D1 may decrease due to aberrant methylation, which is related to the poor prognosis of BC patients.

PRLR is the receptor of prolactin that promotes the growth of breast tissues, and it belongs to the transmembrane receptor of cytokine receptor family [22]. Previous studies have found that PRLR plays an important role in pancreatic cancer [23], ovarian cancer [24], prostate cancer [25], Hodgkin’s lymphoma [26], prolactinoma [27], and BC [28]. PRLR is expressed in most BC cells, but no consensus has been reached regarding whether it is overexpressed in BC tissues. Prospective studies have confirmed that PRLR is overexpressed in ER+ BC [29], which is coincident with our finding that PRLR was overexpressed in BC tissues. Nevertheless, some studies also showed that PRLR was underexpressed in BC tissues [30]. Such discrepancies might be ascribed to the presence of subtypes of PRLR and BC types. This study also found that the abnormal expression of PRLR in BC might be related to mutation and methylation. Additionally, a case-control study by Annika Vaclavicek et al. found the genetic variations of the PRLR gene in BC [31], and Sulggi A Lee et al., by sequencing PRLR, confirmed that a PRLR mutation could occur in BC cells [32]. All these results indicated that PRLR might develop mutations in BC. Furthermore, Li Wang and other researchers confirmed that PRLR had DNA methylation loss, mutations, and increased expression in BC cells [33], which was in line with the finding of this study. In addition, PRLR is mainly involved in JAK2–STAT5, Ras–Raf–MEK–MAPK, Ras–PI3K, Nek3–Vav2–Rac1, and other signaling pathways to regulate cell proliferation, migration, differentiation, and other processes [28,34,35,36,37], thus exerting effects on the development and progression of BC. Given the possible involvement of PRLR in BC, some studies proposed that PRLR could serve as a therapeutic target for BC, and developed PRLR antibody–drug conjugates, such as REGN2878–DM1, antibody targeting hPRLr-ECD, and ABBV-176, among others [37,38,39]. However, these treatments are still in the experimental stage and have not achieved the desired results. Therefore, it is urgent to fully understand how PRLR works mechanistically in BC, thereby providing a new approach to the research of PRLR-related medications for BC.

This study found that PRLR was significantly enriched in cytoskeleton-dependent intracellular transport. Prior studies have shown that cytoskeleton-dependent intracellular transport was related to tumor immunity. ShiYongNeo et al., showed that NK cells could efficiently express and secrete CD73 through vesicle transport, and promote tumor immune escape [40]. Meanwhile, Bastian Krenz et al. demonstrated that MYC and MIZ1 controlled the immune escape of pancreatic ductal adenocarcinoma by inhibiting the vesicular transport of dsRNA [41]. Therefore, we theorized that PRLR might bear a relationship with tumor immunity of BC. Yuexian Zhou et al., proposed that T-cell-mediated cytotoxicity was related to PRLR expression [42] and Tomohiro Yonezawa et al. showed that knocking down long PRLR might inhibit distant metastasis by modifying the tumor microenvironment, and the type of immune infiltrating cells in liver metastasis in the homologous model of 4T1 mouse BC cell line experienced changes [43]. All these results suggested that tumor immune infiltration might partially explain the roles of PRLR in BC.

We used TIMER to evaluate the influence of immune cell infiltration on the clinical prognosis of BC patients, and found that low levels of B cells, CD8+ T cells, CD4+ T cells, neutrophils, and dendritic cells were all related to the poor prognosis of BC patients. However, immune infiltration analysis showed that PRLR expression was feebly correlated with the infiltration level of CD8+ T cells, but not with that of CD4+ T cells, neutrophils, and dendritic cells, indicating that PRLR might affect the prognosis of BC patients but not by regulating the abundance of tumor immune cells. Therefore, immune cell infiltration may not be the cause of poor prognosis in BC patients with high PRLR expression.

The abundance of infiltrating immune cells can exert a substantial impact on the prognosis of BC patients and tumor cells can also evade the identification and clearance by the immune system through immune checkpoints, both contributing to the prognosis of BC. Furthermore, this study evaluated the correlation between some common immune checkpoints and the prognosis of BC patients, and the relationship between PRLR and these immune checkpoints. The results showed that BC patients with a high expression of PDCD1, CTLA4, TIGIT, and BTLA had a better clinical prognosis, while the expression of PRLR was negatively correlated with the expression of these immune checkpoints, indicating that PRLR might participate in the immune process of BC by inhibiting the expression of the above immune checkpoints, and targeting PRLR might improve the efficacy of BC immunotherapy. To sum up, this study suggested that PRLR might affect the prognosis of BC patients by regulating the expression of immune checkpoints.

CACNA2D1 is a gene encoding a T-type calcium channel subunit α2δ1. CACNA2D1 is involved in the regulation of calcium current density and dynamics of calcium channels, and plays an important role in excitation–contraction coupling. It was found to be profoundly involved in the metabolism, immune system, cardiovascular system, musculature, and other phenotypes in vivo [44]. In line with these findings, this study showed that it was functionally implicated in myofibril formation, adhesive spots, and the development of muscular structure. Prior studies found that CACNA2D1 played a certain role in the carcinogenesis of prostate cancer [45], bladder cancer [46], epithelial ovarian cancer [47], laryngeal squamous cell cancer [48], gastric cancer [49], non-small-cell cancer [50], and other cancers, but very few studies examined the association between CACNA2D1 and BC. Meng Li et al. showed that the cells overexpressing α2δ1 were involved in the self-renewal, differentiation, invasion, and metastasis of the tumor. They believed that CACNA2D1 might potentially act as a biomarker of BC stem cells and might provide a new target for the treatment of BC [51], as suggested by this study. This study revealed that the expression of CACNA2D1 was low in BC tissues, and the patients with low expression had poor prognosis. Additionally, the methylation level of CACNA2D1 was high in BC tissues, and one methylation site (cg25161868) was in the 3’UTR region and, at the same time, in the open-sea. This abnormal methylation might be responsible for the low expression of CACNA2D1 in BC.

Functionally, CACNA2D1 was found to be mainly involved in myofibrillar composition, adhesive spots, and muscular structure development, and cytoskeleton-dependent intracellular transport and the cytoskeleton were all related to tumor immunity. Therefore, we were led to assume that CACNA2D1 might be related to tumor immunity. However, so far, no studies have previously examined the relationship between CACNA2D1 and BC immunity. In this study, analyses on immune infiltration and immune checkpoints of CACNA2D1 found that CACNA2D1 was positively correlated with immune cells such as B cells, CD8+ T cells, CD4+ T cells, neutrophils, and dendritic cells, and the low CACNA2D1 expression in the immune cells was related to the poor prognosis of BC, suggesting that CACNA2D1 might impact the prognosis of BC patients by regulating the abundance of tumor immune cells. In addition, CACNA2D1 was positively correlated with the expression of immune checkpoints, such as CTLA4, TIGIT, and BTLA, which, in turn, are positively correlated with the prognosis of BC patients. These findings indicated that CACNA2D1 might influence the prognosis of BC patients by regulating the expression of immune checkpoints, and targeting CACNA2D1 may improve the efficacy of immunotherapy in BC.

This study has some limitations. First, BC types were not taken into consideration in the grouping of the samples. Second, only TCGA database was used in the model construction and verification, and no external data sets were used for verification. In addition, the expression and functions of the selected molecules were not experimentally confirmed. Further studies are warranted to verify the reliability and accuracy of the model. Moreover, in vitro and in vivo experiments are needed to substantiate the roles of PRLR and CACNA2D1 in the pathogenesis of BC and their effects on the prognosis of BC.

To sum up, this study established a model that evaluated BC prognosis on the basis of the PTEN-related ceRNA network, and found that PRLR and CACNA2D1 might be involved in BC by regulating the abundance of tumor immune infiltrating cells and the expression of immune checkpoints, thus affecting the prognosis of BC patients. PRLR and CACNA2D1 hold promise to serve as therapeutic targets for BC.

## Figures and Tables

**Figure 1 jpm-12-02086-f001:**
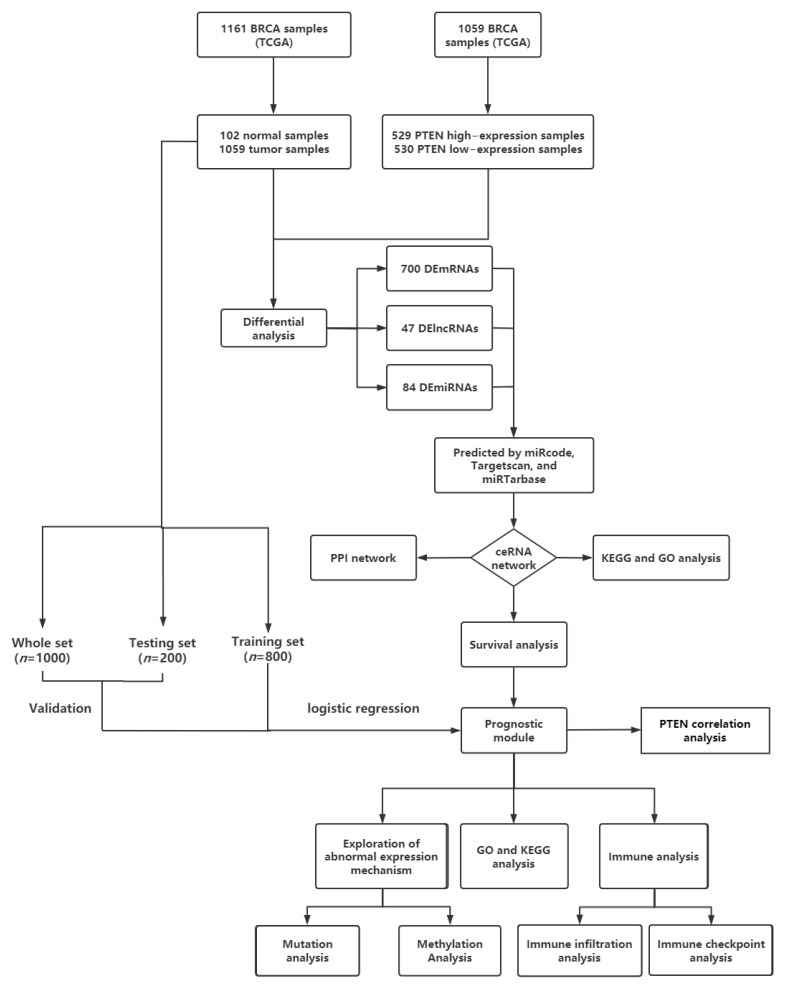
Flowchart of construction and analysis of ceRNA network.

**Figure 2 jpm-12-02086-f002:**
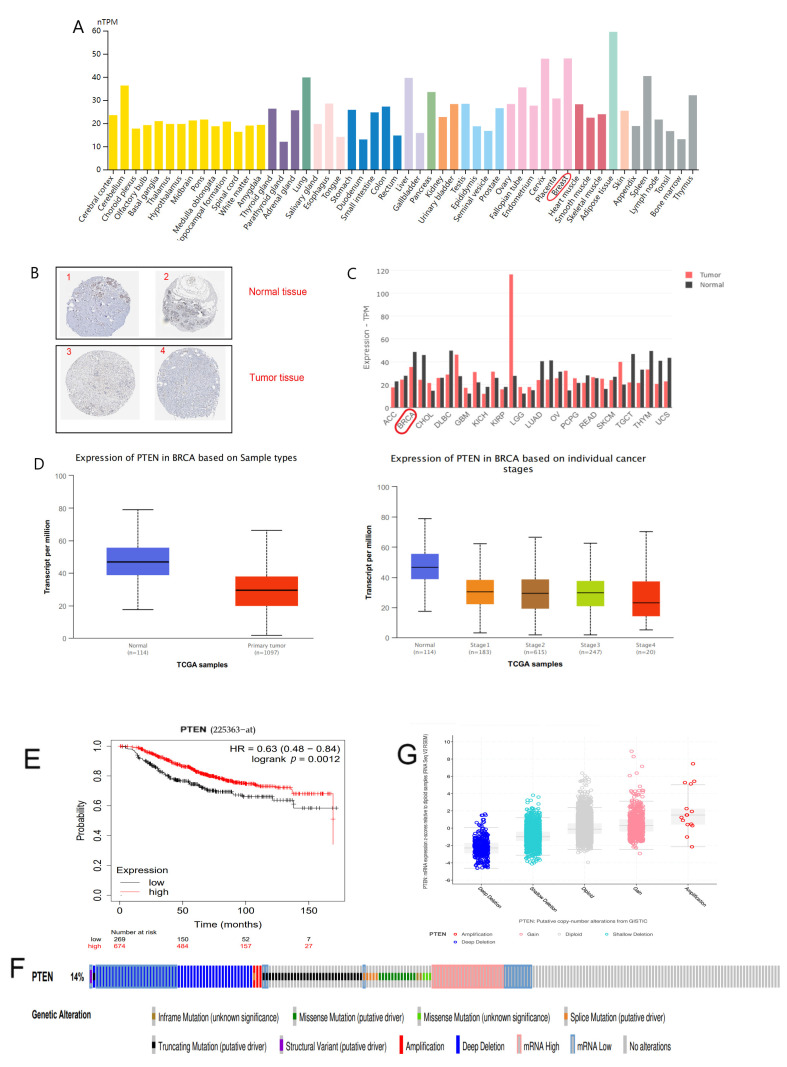
(**A**) Distribution of PTEN expression in pan-cancer tissues. (**B**) Results of immunohistochemical staining of PTEN in normal breast tissue and tumorous breast tissues as shown by HPA database. (**C**) Expression of PTEN in normal tissues and tumor tissues as revealed by GEPIA2. (**D**) Expression of PTEN in terms of BC samples and clinical stages as exhibited by UALCAN. (**E**) The KM survival curve of PTEN by Kaplan–Meier plotter. (**F**) The distribution of PTEN genomic alterations in TCGA BRCA presented on a OncoPrint plot. (**G**) The association between PTEN copy number and mRNA expression as shown in a dot plot generated by cBioPortal.

**Figure 3 jpm-12-02086-f003:**
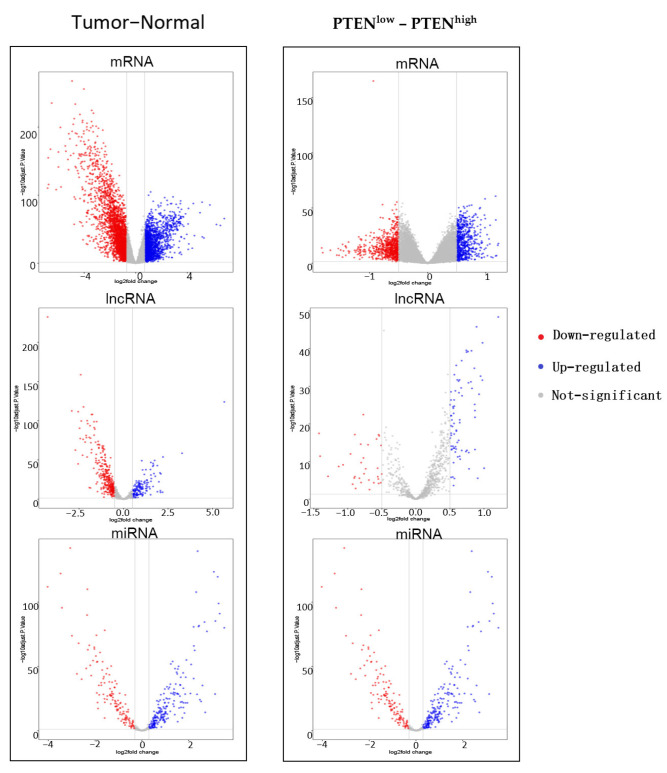
Volcano plots of demRNAs, delncRNAs, and demiRNAs between the expression of tumor and normal group as well as PTEN^low^ and PTEN^high^ group: Red denotes down-regulated RNAs and blue represents up-regulated RNAs.

**Figure 4 jpm-12-02086-f004:**
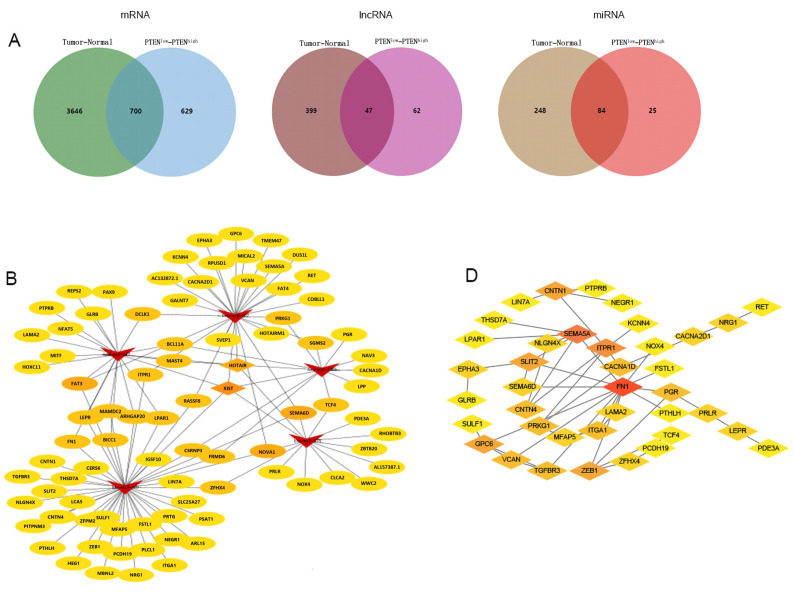
(**A**) Venn diagrams of pooled demRNAs, delncRNAs, and demiRNAs. (**B**) The regulatory ceRNA network in BC (arrows represent miRNAs, diamonds denote lncRNAs, and ovals are mRNAs. (**C**) Functional enrichment analysis of the mRNAs in the ceRNA network. (**D**) The PPI network of the mRNAs in the ceRNA network.

**Figure 5 jpm-12-02086-f005:**
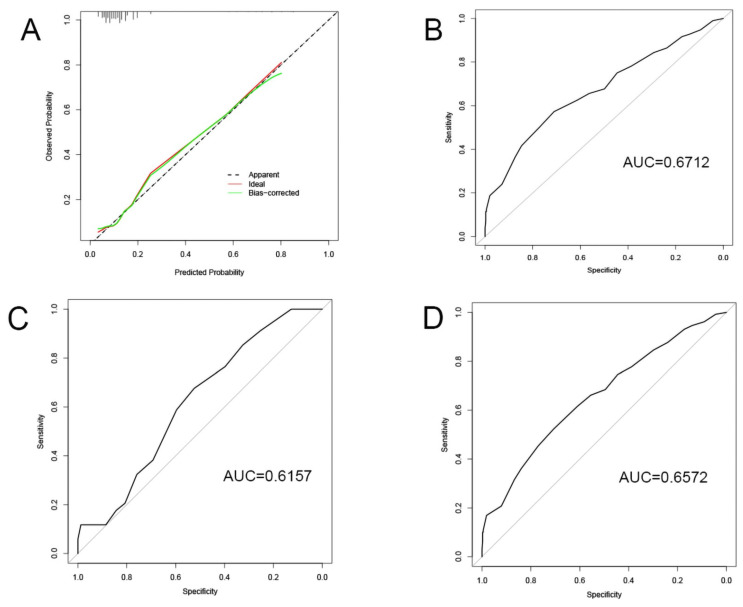
Validation of the model: (**A**) Calibration curve; (**B**–**D**) ROC curves of training set, testing set, and whole set.

**Figure 6 jpm-12-02086-f006:**
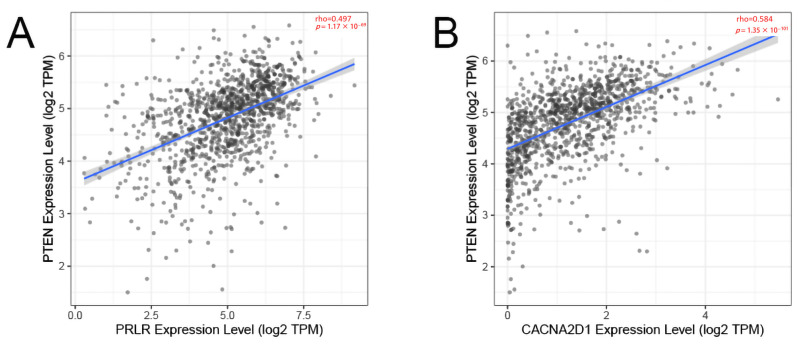
The correlation between PRLR and PTEN (**A**) and the correlation between CACNA2D1 and PTEN (**B**).

**Figure 7 jpm-12-02086-f007:**
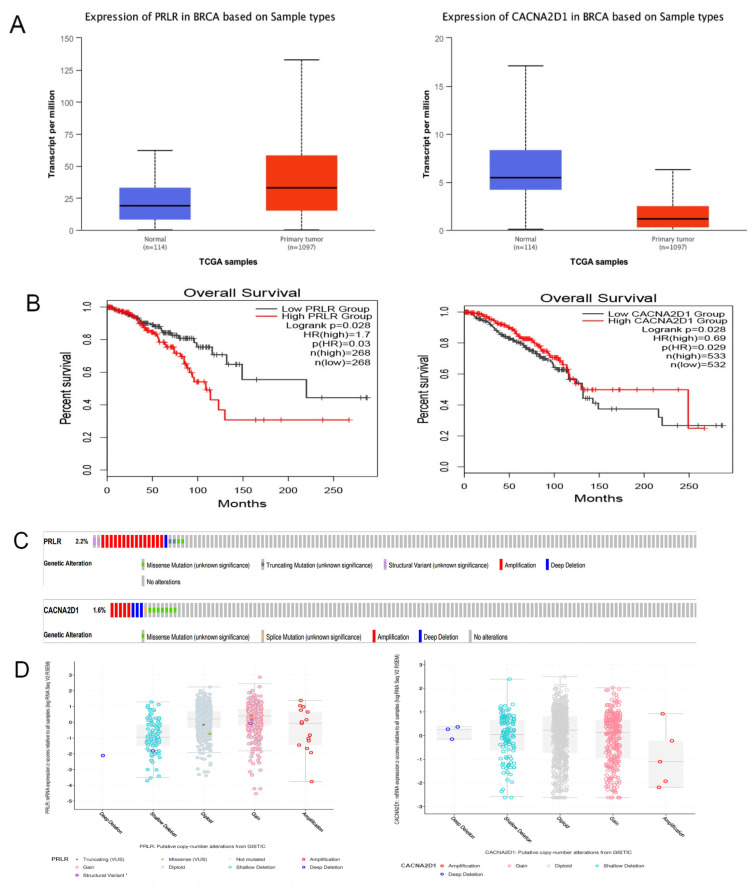
(**A**) Expression of PRLR and CACNA2D1 in BC tissue and normal tissue by using UALCAN. (**B**) The overall survival of PRLR and CACNA2D1 in BC by using GEPIA2. (**C**) The distribution of PRLR and CACNA2D1 genomic alterations in TCGA BRCA shown on a cBioPortal OncoPrint plot. (**D**,**E**) The association between PRLR and CACNA2D1 copy number and mRNA expression displayed in a dot plot (**D**) and correlation plot (**E**) generated by cBioPortal.

**Figure 8 jpm-12-02086-f008:**
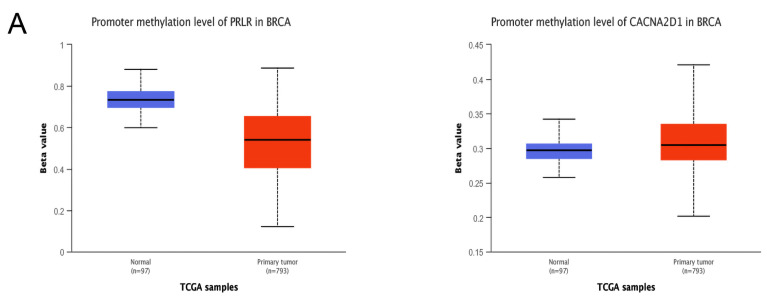
Methylation analysis of CACNA2D1: (**A**) Methylation of CACNA2D1 in BRCA was evaluated by UALCAN. (**B**,**C**) The methylation sites of PRLR and CACNA2D1 DNA sequence association with gene expression were visualized by MEXPRESS. The expression of them was illustrated by the blue line in the center of the plot. Pearson’s correlation coefficients and *p* values for methylation sites and query gene expression are shown on the right side.

**Figure 9 jpm-12-02086-f009:**
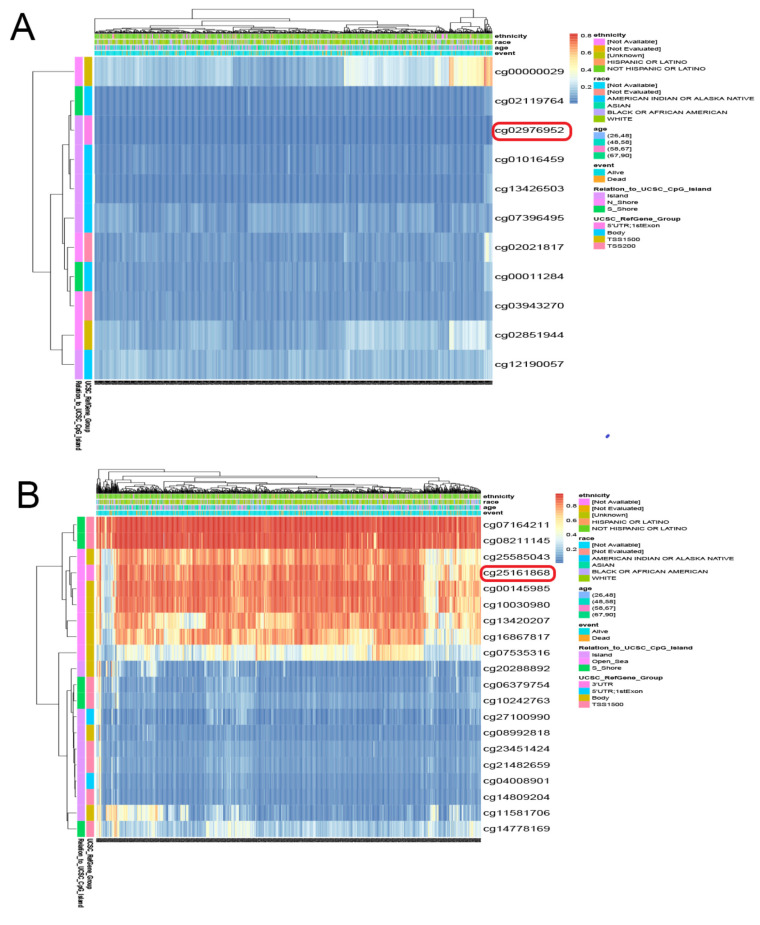
Heatmaps of different methylated regions associated with PRLR (**A**) and CACNA2D1 (**B**) generated by MethSurv.

**Figure 10 jpm-12-02086-f010:**
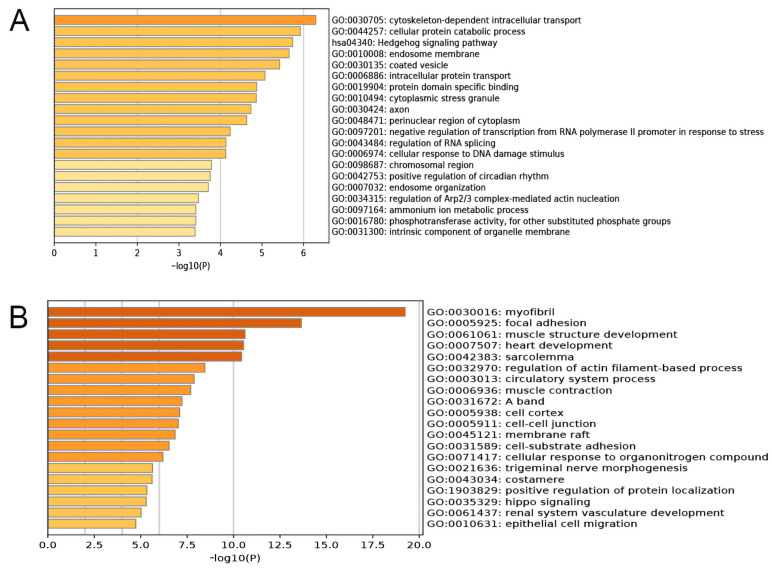
Functional enrichment analysis (including GO and KEGG) of PRLR (**A**) and CACNA2D1 (**B**) associated genes in BRCA.

**Figure 11 jpm-12-02086-f011:**
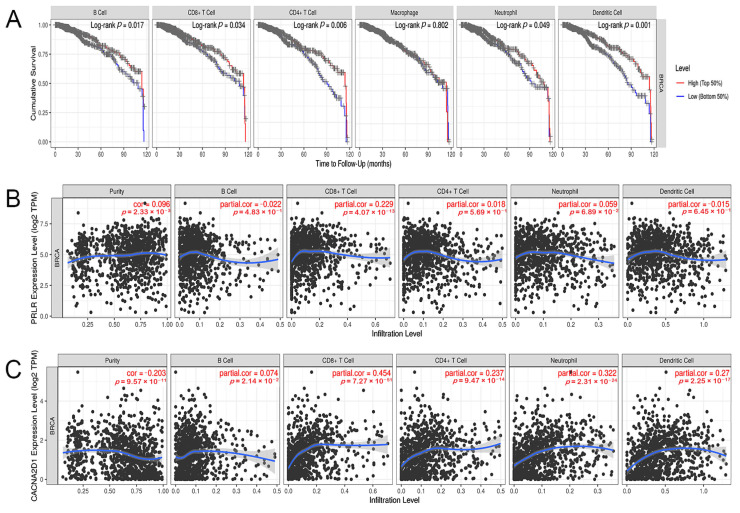
(**A**) Kaplan–Meier plots were used to analyze the immune infiltration and overall survival rate of BRCA. (**B**) Correlation of PRLR expression with immune infiltration level in BRCA. (**C**) Correlation of CACNA2D1 expression with immune infiltration level in BRCA.

**Figure 12 jpm-12-02086-f012:**
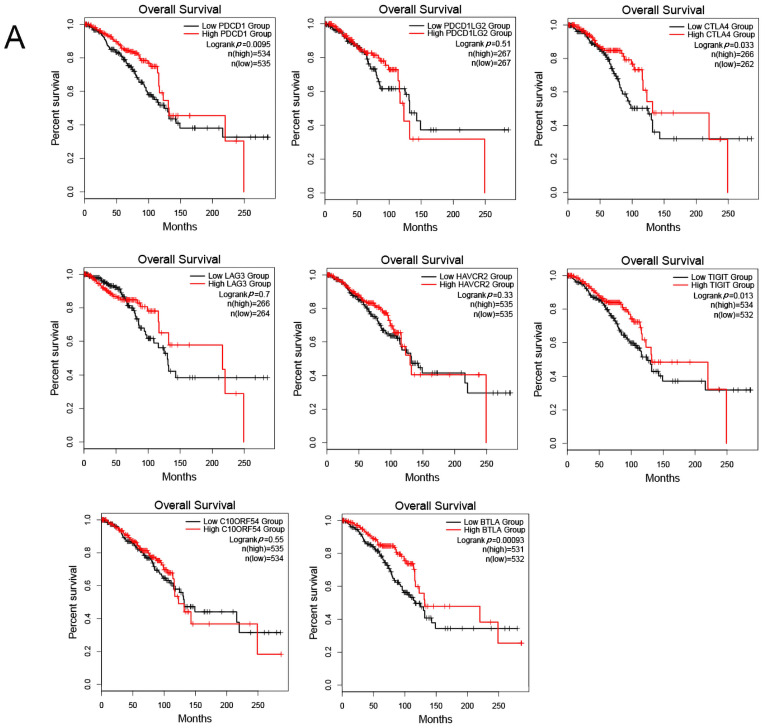
(**A**) Kaplan–Meier plots were used to analyze the immune checkpoints and overall survival rate of BRCA. (**B**) Relationship between immune checkpoints and PRLR expression. (**C**) Relationship between immune checkpoints and CACNA2D1 expression.

**Table 1 jpm-12-02086-t001:** Findings of immunohistochemical staining of PTEN in BC in HPA database.

Number	Tissue Type	ID	Age	Gender	Staining
1	Normal	3856	27	Female	Low
2	Normal	2042	75	Female	Low
3	Tumor	2252	47	Female	Not detected
4	Tumor	2199	60	Female	Not detected

**Table 2 jpm-12-02086-t002:** Variables and their parameters in the model.

Variable	B	Wald	OR with CI	*p* Value
Intercept	−2.293	35.349	0.01 (0.046–0.211)	<0.001
M	3.182	27.803	24.083 (7.941–89.816)	<0.001
N	0.627	6.869	1.871 (1.179–3.019)	0.009
CACNA2D1	−0.470	9.247	0.625 (0.459–0.843)	0.002
PRLR	0.352	5.469	1.421 (1.061–1.915)	0.019

**Table 3 jpm-12-02086-t003:** Chi-square test of the model.

Variable	Df	Pr (>Chi)
M	1	<0.001
N	1	0.011
CACNA2D1	1	0.014
PRLR	1	0.018

**Table 4 jpm-12-02086-t004:** Expression of PRLR and CACNA2D1 in BC tissue and normal tissue.

Data Set	Symbol	LogFC	*p* Value	Expression
GSE21422	PRLR	3.34826808	1.16 × 10^−3^	Up
	CACNA2D1	−4.18492567	1.34 × 10^−4^	Down
TCGA	PRLR	1.335680424	2.52 × 10^−21^	Up
	CACNA2D1	−1.93331686	1.13 × 10^−75^	Down

## Data Availability

The data presented in this study are publicly available in the TCGA database, GEO database, and HPA database, and the analysis and results presented in this work do not infringe copyright.

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
