# Peer review of "PRLR and CACNA2D1 Impact the Prognosis of Breast Cancer by Regulating Tumor Immunity"

_jpm, 2022, doi:10.3390/jpm12122086_

Round 1
Reviewer 1 Report
Review for “PRLR and CACNA2D1 impact the prognosis of breast cancer by regulating tumor immunity”
The study is cleverly designed and continues the previous research of this group. The manuscript is well-written and easy to follow for even the researchers outside of this field of study. The introduction provides enough background and needs no additional references. Research is appropriate and well-designed, and methods are adequate and in line with the latest research trends. However, some results are not so clearly presented, and several conclusions do not seem to be, or are insufficiently, based on the presented results.
For example, the Authors state in Figure 8 that they found 15 methylation sites of CACNA2D1 which were positively correlated with its expression, but this is not so clear from Figure 8. Also, on the right side of Figure 8B, we see 16, and not 15, positively correlated methylation sites. Can the Authors explain how they came up with the number 15? Furthermore, in Figure 8, the Authors provide detailed methylation analysis of CACNA2D1 but it is unclear why they didn’t do the same for PRLR methylation as well. This brings into question their claim in the Discussion that “PRLR may be overexpressed in BC tissues through mutation and hypomethylation, thus affecting the prognosis of BC patients”. Can the Authors elaborate and provide more data/explanations on this?
Lastly, I would suggest drawing a graphical abstract which could help visualize to the layman audience the main idea and results of the presented research model.
The following is the list of some additional minor issues that need to be addressed before the manuscript is, in my opinion, suitable for publication.
· Please provide the full explanation for all abbreviations used in the text such as FCGR1A, TRRAP, PTEN, PPI, PRLR and CACNA2D1
· Please check for misspelled abbreviations such as UACLAN instead of UALCAN
· Regarding the captions for Figure 3. Check if the red color really denotes up-regulated RNAs because the legend of the figure states otherwise (red: down-regulated). Please elaborate on this and correct if this is a mistake
Author Response
Dear reviewer,
Thank you for your comments on our manuscript entitled “PRLR and CACNA2D1 impact the prognosis of breast cancer by regulating tumor immunity” (ID: jpm-2070132). Those comments are very valuable and helpful for revising our paper, as well as the important guiding significance to our research. We have studied comments carefully and have made correction which we hope meet with approval. Revised portions are marked in red in the paper. The responses to the reviewer's comments are marked in red and presented following:
Point1: the Authors state in Figure 8 that they found 15 methylation sites of CACNA2D1 which were positively correlated with its expression, but this is not so clear from Figure 8. Also, on the right side of Figure 8B, we see 16, and not 15, positively correlated methylation sites. Can the Authors explain how they came up with the number 15?
Response 1: We carefully count the number of CACNA2D1’s methylation sites this time, and the number is 16 instead of 15. We are sorry to make such a mistake and have corrected it in the paper.
Point 2:Furthermore, in Figure 8, the Authors provide detailed methylation analysis of CACNA2D1, but it is unclear why they didn’t do the same for PRLR methylation as well. This brings into question their claim in the Discussion that “PRLR may be overexpressed in BC tissues through mutation and hypomethylation, thus affecting the prognosis of BC patients”. Can the Authors elaborate and provide more data/explanations on this?
Response 2: We are extremely grateful to you for pointing out this problem. We did analyze the methylation of PRLR at the first time and didn’t show the detailed results with an idea that the methylation of PRLR was low in breast cancer so there may be no need to show more details. However, after discussion, we agree with the comment and add these data in the revised manuscript.
Point 3: I would suggest drawing a graphical abstract which could help visualize to the layman audience the main idea and results of the presented research model.
Response 3: Thank you for your suggestion, and we have drawn a graphical abstract in the revised manuscript.
Point 4: Please provide the full explanation for all abbreviations used in the text such as FCGR1A, TRRAP, PTEN, PPI, PRLR and CACNA2D1
Response 4: We deeply appreciate your suggestion. We have provided the full explanation for all abbreviations used in the text such as FCGR1A, TRRAP, PTEN, PPI, PRLR and CACNA2D1.
Point 5: Please check for misspelled abbreviations such as UACLAN instead of UALCAN
Response 5: The correct spelling of this word is UALCAN, and we have corrected the misspelled abbreviations in the text.
Point 6: Regarding the captions for Figure 3. Check if the red color really denotes up-regulated RNAs because the legend of the figure states otherwise (red: down-regulated). Please elaborate on this and correct if this is a mistake.
Response 6: Thank you so much for pointing out the mistake. The red color denotes down-regulated RNAs, and we have corrected this mistake in the legend of the figure 3.
We would love to thank you for allowing us to resubmit a revised copy of the manuscript and we highly appreciate your time and consideration. We wish good health to you, your family, and community. Your careful review has helped to make our study clearer and more comprehensive.

Reviewer 2 Report
Can the authors comment on hot tumors and cold tumors in relation to PRLR and CACNA2D1 ?
Author Response
Thank you for your comment on our manuscript entitled “PRLR and CACNA2D1 impact the prognosis of breast cancer by regulating tumor immunity” (ID: jpm-2070132). The comment is very valuable and has the important guiding significance to our research. We have studied the comment carefully. The response to your comment is marked in red and presented following:
Point 1: Can the authors comment on hot tumors and cold tumors in relation to PRLR and CACNA2D1?
Response 1: We are grateful for the suggestion. We looked up the relevant articles of hot tumors and cold tumors and then found that different types of breast cancer have different proportions of infiltrating lymphocytes. More studies are needed to comment clearly on hot tumors and cold tumors in relation to PRLR and CACNA2D1, including the analysis of the proportion of infiltrating lymphocytes in different subtypes of breast cancer, the expression of PRLR and CACNA2D1 in different subtypes of breast cancer and in lymphocytes, and so on. Due to the length of our paper at present and our research plan, we hope to incorporate these contents into the next research plan. Thank you again for pointing out the direction for our next research.
Thank you for your suggestion for the manuscript and we highly appreciate your time and consideration. We wish good health to you, your family, and community. Your careful review has helped to make our next study clearer and more comprehensive.

Reviewer 3 Report
The work is devoted to the bioinformatic search for potential targets for breast cancer therapy and to establish the role of PRLR and CACNA2D1 in the development and progression of breast cancer. Despite the fact that the authors used only bioinformatics methods, they succeeded, using a well-chosen set of approaches, to identify a number of interesting patterns that can be of significant practical importance.
The text needs minor editing to remove some typos. For example, on line 14 a space is missing. The illustrative material looks a bit overloaded, you should think about how to make it more accessible. This applies, in particular, to Figure 8. Perhaps some of the data should be placed in the supplementary, if possible. Also, in my opinion, it is incorrect to present the immunohistochemical data obtained by other researchers in the “Results” section, especially without reference to the original source (Figure 2B).
Nevertheless, the article represents the result of interesting work and, after a slight revision, can be published.
Author Response
Dear reviewer,
Thank you for your comments on our manuscript entitled “PRLR and CACNA2D1 impact the prognosis of breast cancer by regulating tumor immunity” (ID: jpm-2070132). Those comments are very valuable and helpful for revising our paper, as well as the important guiding significance to our research. We have studied comments carefully and have made correction which we hope meet with approval. Revised portions are marked in red in the paper. The responses to the reviewer's comments are marked in red and presented following:
Point 1: The text needs minor editing to remove some typos. For example, on line 14 a space is missing.
Response 1: Thank you so much for pointing out the mistakes. And we have corrected them in the revised manuscript.
Point 2: The illustrative material looks a bit overloaded; you should think about how to make it more accessible. This applies, in particular, to Figure 8. Perhaps some of the data should be placed in the supplementary, if possible.
Response 2: We deeply appreciate your suggestion.
Point 3: Also, in my opinion, it is incorrect to present the immunohistochemical data obtained by other researchers in the “Results” section, especially without reference to the original source (Figure 2B).
Response 3: We are grateful for the suggestion. The immunohistochemical data obtained by other researchers in the “Results” section is to illustrate the gene expression from the protein level. And the data is obtained from the HPA database which is a public database. According to your advice, it has been emphasized in the method part, body and "Data Availability Statement" that the immunohistochemical data was acquired from public database. And we also cited the website and article according to the HPA database.
Our deepest gratitude goes to you for your careful work and thoughtful suggestions that have helped improve this paper substantially.
